Effects of monocropping soil on plant growth and rhizosphere microbial community structure of Salvia miltiorrhiza Bge

Ju Ji Dong 1 2 3
Meng Yuan 4
Zhou Bingqian 1 2
Yang Guohong 1 2
Fu Xinyu 1 2 3
Liu Wei 1 2 liuwei0074@163.com
1 Key Laboratory for Applied Technology of Sophisticated Analytical Instruments of Shandong Province, Shandong Analysis and Test Center, Qilu University of Technology (Shandong Academy of Sciences) , Jinan , China
2 Key Laboratory for Natural Active Pharmaceutical Constituents Research in Universities of Shandong Province, School of Pharmaceutical Sciences, Qilu University of Technology (Shandong Academy of Sciences) , Jinan , China
3 Shandong University of Traditional Chinese Medicine , Jinan , China
4 Dongying Hospital of Traditional Chinese Medicine (Dongying Shengli Hospital) , Dongying , China
García-Contreras Rodolfo
Electronic publication date: 2025 Nov 28
Publication date: 2025
Volume: 13
Electronic Location ID: e20379
Received 2025 Apr 29; Accepted 2025 Oct 21
Copyright: © 2025 Ju et al.
Copyright year: 2025
Copyright holder: Ju et al.
License: This is an open access article distributed under the terms of the Creative Commons Attribution License, which permits unrestricted use, distribution, reproduction and adaptation in any medium and for any purpose provided that it is properly attributed. For attribution, the original author(s), title, publication source (PeerJ) and either DOI or URL of the article must be cited.
License URL: https://creativecommons.org/licenses/by/4.0/

Keywords: Salvia miltiorrhiza, Rhizosphere microorganisms, Monocropping obstacle, High-throughput sequencing technology

Funding: National Natural Science Foundation of China 82173917 National Key Research and Development Plan 2023YFC3503801 China Agriculture Research System of MOF and MARA CARS-21 Qilu University of Technology (Shandong Academy of Sciences) Science, Education and Industry Integration In-novation Pilot Project 2023PYI004 This work was supported by the National Natural Science Foundation of China (Grant No. 82173917), National key research and development plan (2023YFC3503801), China Agriculture Research System of MOF and MARA (CARS-21) and Qilu University of Technology (Shandong Academy of Sciences) Science, Education and Industry Integration In-novation Pilot Project (Grant No. 2023PYI004). The funders had no role in study design, data collection and analysis, decision to publish, or preparation of the manuscript.

==============================
Salvia miltiorrhiza (Danshen) is a commonly utilized remedy for various blood stasis syndromes, cardiovascular, and cerebrovascular diseases. The practice of continuous cropping presents a notable challenge to the production of high-quality S. miltiorrhiza due to the lack of rotation or soil remediation. Despite this, the precise mechanism, particularly the impact of alterations in the rhizosphere microbial community structure on the development of obstacles related to continuous cropping, remains obscure. The constitution of the rhizosphere microbial community plays a pivotal role in plant growth and has the potential to exacerbate issues associated with continuous cropping. This study aimed to comprehensively compare the impact of continuous vs. non-continuous cropping soils on the growth, physiological and biochemical characteristics, accumulation of active ingredients, and rhizosphere microbial community structure of S. miltiorrhiza seedlings to elucidate the microbial ecological mechanism behind continuous cropping challenges. A pot experiment was executed to assess the effects of continuous cropping on the morphological and physiological parameters of S. miltiorrhiza. High-throughput sequencing technology utilizing the NovaSeq platform was employed to sequence and analyze the V4 region of bacterial 16S rDNA and the ITS1 region of fungi in the rhizosphere. The findings revealed that continuous cropping soil led to a reduction in S. miltiorrhiza biomass, manifesting as wilting and stunted growth, diminished effective leaf area, main root length and diameter, reduced levels of total chlorophyll, carbohydrates, and active ingredients, as well as compromised photosynthesis. In the rhizosphere soil, both the composition and function of the fungal community have undergone significant changes, and the fungal diversity has shown a clear increasing trend; in contrast, the change in bacterial diversity is very slight. In conclusion, continuous cropping soil substantially hinders the growth of S. miltiorrhiza, diminishes its physiological functions, and disrupts the structure of the rhizosphere microbial community. These changes likely contribute significantly to the exacerbation of challenges associated with continuous cropping. This investigation furnishes a theoretical foundation for comprehending the microecological mechanism underpinning continuous cropping challenges in S. miltiorrhiza and devising strategies for soil amelioration.

Introduction

The dried roots and rhizomes of S. miltiorrhiza Bge. exhibit diverse pharmacological effects that offer health benefits to humans. These effects include anti-inflammatory and antioxidant properties, promotion of blood circulation, and significant contributions to the prevention and treatment of cardiovascular and cerebrovascular diseases (Chang et al., 2016). Consequently, S. miltiorrhiza is increasingly recognized as a valuable botanical resource in pharmaceuticals, functional foods, skincare, cosmetics, and other related industries (Nitrayová et al., 2014; Wang et al., 2017). The rising demand for S. miltiorrhiza has led to the practice of monocropping. However, monocropping can result in soil degradation, increased susceptibility to pests and diseases, and higher rates of seedling mortality. These issues have emerged as key factors impeding the enhancement of yield, quality, and market expansion of S. miltiorrhiza (Liu et al., 2020; Zhang, Sun & Ye, 2005; Zeeshan et al., 2023; Li & Zhang, 2015).

Medicinal plant quality assessment involves morphological and physicochemical property evaluations. Photosynthesis drives morphological development in green plants, utilizing carbohydrates as energy sources for growth (Wu et al., 2025). The synthesis and accumulation of active compounds in medicinal plants are often linked to photosynthetic products (Zhang et al., 2021). However, many medicinal rhizomatous plants, including Pseudostellaria heterophylla, Lilium brownii, Atractylodes macrocephala, and Panax ginseng, are susceptible to quality decline and crop failure due to monocropping (Wu et al., 2015a; Han et al., 2024; Wu et al., 2019, 2015b). The S. miltiorrhiza industry also experiences yield and quality changes from monocropping effects (Zhao et al., 2018). Hence, a systematic assessment of the impact of continuous cropping on the above- and below-ground morphology, biomass accumulation, physiological metabolism, and active ingredient synthesis of S. miltiorrhiza Bunge plants is crucial for elucidating the mechanisms underlying continuous cropping obstacles.

Soil microorganisms play a crucial role in nutrient transformation and the breakdown of harmful substances, which are closely associated with plant growth and metabolism. Plants, soil, and microorganisms engage in material exchange and signal transmission within the rhizosphere environment, collectively forming a stable ecosystem that enhances plant support and protection during stressful conditions, thereby improving their resistance and resilience (Corato, 2020). Continuous cropping can profoundly impact the composition of the rhizosphere microbial community, resulting in an increase in soil-borne pathogens and a decrease in beneficial microorganisms (Pervaiz et al., 2020; Wang et al., 2020; Rousk et al., 2010; Escudero-Martinez & Bulgarelli, 2019). This alteration can escalate crop diseases and impede productivity (Huang et al., 2019; Luo et al., 2019; Liu et al., 2021). Previous research indicates that challenges associated with continuous cropping primarily arise from disruptions in the microbial community, allelopathic autotoxicity, and changes in soil physical and chemical characteristics (Zhao et al., 2018; Xin et al., 2018; Na et al., 2021; Tan et al., 2021). However, the specific changes in the structure of the rhizosphere microbial community and its physiological and ecological consequences in the context of the S. miltiorrhiza continuous cropping system remain inadequately understood.

Pot experiments were conducted to assess the overall impact of soil subjected to continuous cropping challenges on the key physiological processes, secondary metabolite synthesis, and rhizosphere microecological balance of S. miltiorrhiza seedlings in comparison to non-continuous cropping soil. The composition and diversity of bacterial and fungal communities in the rhizosphere soil of S. miltiorrhiza were examined using advanced high-throughput sequencing technology. This research offers a theoretical and practical foundation for investigating the development of continuous cropping obstacles in Chinese medicinal plants and enhancing soil quality to enhance the yield of Chinese medicinal materials.

Materials and Methods

Soil collection and characterization

The experimental soils were collected from adjacent plots in Laiwu Ziguang Ecological Garden, Jinan, Shandong. Both soils shared the same soil type and basic agronomic management to minimize confounding factors. The tested soil was sandy loam, with the following basic physical and chemical properties: calcium 14,890 mg/kg, magnesium 11,030 mg/kg, iron 49,860 mg/kg, manganese 512 mg/kg, copper 53 mg/kg, zinc 135 mg/kg, available phosphorus 34.9 mg/kg, available potassium 283 mg/kg, organic matter 0.75%, and total nitrogen 0.06%. Monocropped soil (Group MS): Collected from a field that had been continuously cropped with S. miltiorrhiza for 2 years, exhibiting typical replant disease symptoms. Non-continuous cropping soil (Group NS): collected from plots in the region where S. miltiorrhiza has not been planted. A comprehensive analysis of the physicochemical properties of these soils, confirming significant degradation in the monocropped soil, has been previously published (Bingqian et al., 2019; Wei et al., 2019). This pre-existing difference in soil state constitutes the baseline for the present pot experiment.

Early cultivation of S. Miltiorrhiza seedlings

On December 14, 2021, seeds were thoroughly washed in pure water, and shriveled and defective seeds were removed. After soaking for 4–5 h, they were sealed and stored in a refrigerator at 4 °C for 7 days to induce dormancy. On the 21st, the sterilized field soil, organic nutrient soil and vermiculite, which were mixed evenly at a ratio of 5:3:2, were spread on the seedling tray. A total of 3–5 seeds were sown in each cell of the tray and then placed in a well-lit area at 26 °C for germination. On December 28, 2021, seedlings displaying leaf deformities and thin stems post-germination were selectively pruned. In March 2022, robust and uniformly sized seedlings were chosen and transferred to flowerpots measuring 15 cm in height and 13 cm in upper diameter. The cultivation trial took place in the illuminated growth chamber at the Shandong Analysis and Test Center, maintaining a diurnal temperature of 26 °C, a nocturnal temperature of 18 °C, and a 12-h light-dark cycle. Disease incidence was closely monitored throughout the experiment.

Experimental grouping and subsequent treatment

The experiment was divided into two groups, each containing five biological replicates (five pots of S. miltiorrhiza per group). The pots filled with non-monocropping soil were designated as Group NS, and those filled with monocropping soil were labeled as Group MS (see Table 1 for soil information). In October 2022, when the S. miltiorrhiza plants in the monocropping soil showed disease symptoms, the plants were collected for morphological and physiological activity determination. The rhizosphere soil samples (fresh soil at a depth of approximately 1–2 mm from the root surface) of both Group NS and Group MS were collected by means of the root shaking technique (Guo et al., 2020). Subsequently, the soil samples were sieved through a 40-mesh sieve (with a pore diameter of 0.425 mm), and then stored in dry ice prior to being transported to Novogene Technology Co., Ltd. (located in Beijing, China).

Table 1 Relevant information about potting soil Group NS and Group MS.

Treatment	Sterilized soil	Non-monocropping soil	Monocropping soil	
NS group	50%	50%	0	
MS group	50%	0	50%	
Note:

The non-monocropping soil is the field soil that has not been planted with S. miltiorrhiza, and the monocropping soil is the field soil that has been continuously planted with S. miltiorrhiza for two years or more in the same area

Determination of morphological indices

The S. miltiorrhiza plants in groups F and L were taken out from a ceramic flowerpots, and the soil was shaken off (Ju et al., 2024). After cleaning, the fresh weights (g) of the aboveground and underground plant parts were measured. Then, plants of S. miltiorrhiza were heated in an oven at 85 °C for 20 min, cooled to 70 °C, and dried to a constant weight. The dry weight data were recorded and the drying rate was calculated. The number of complete leaves, number of dry or diseased leaves, and total number of leaves in each group were recorded. The maximum leaf length and maximum leaf width of each leaf were measured, and the leaf length: leaf width ratio and leaf area (in cm2) were calculated (Ju et al., 2024). The length and cross-sectional diameter of the main root of S. miltiorrhiza from the reed head to the root tip were determined, which were recorded as the longest root length and the main root diameter, respectively (in cm). Furthermore, the number of roots (strip) with a diameter >0.2 cm was recorded. One-way analysis of variance and Tukey test were used to calculate whether there were significant differences among the treatment groups (Ju et al., 2024).

Portions of this text were previously published as part of a preprint (DOI: 10.21203/rs.3.rs-4565313/v1; Ju et al., 2024).

Evaluation of physiological indices

Determination of chlorophyll and sugar contents

The fresh leaves of groups NS and MS were collected and their veins were removed, and cut into fragments of 1–2-mm width (Ju et al., 2024). Then, 1.0 g of the leaves fragments were immersed in 50 mL of 95% ethanol for extraction (dark) until the leaf tissue became completely white (Yixuan, Yinhong & Lujing, 2024). The absorbance (A) of the extract was determined using UV-2700 ultraviolet spectrophotometer at the wavelengths of 665, 649, and 470 nm, and the contents of total chlorophyll, chlorophyll a (C-a), chlorophyll b (C-b), and carotenoids (C-carotenoid) were calculated as follows:

C-a=13.95×A665-6.88×A649.

C-b=24.96×A649-7.32×A665.

C-carotenoid=(1000×A470-2.05×C-a-114.8×C-b)/24.

Chlorophyll content (mg/g) = (C chlorophyll × V extract × dilution)/sample fresh weight (Ju et al., 2024).

To determine the soluble sugar and sucrose contents, 3.0 g of the leaves and roots of S. miltiorrhiza were extracted with ethanol (80%) in a water bath (80 °C) for 30 min. After cooling to room temperature, the extracts were centrifuged (4,000 rpm for 1.5 min) and the supernatants were collected. The soluble sugar content was measured using anthrone colorimetry, and the sucrose content was determined by resorcinol method (Zhang et al., 2023; Monsigny, Petit & Roche, 1988). For the calculation of glucose and fructose contents, the leaves and roots of S. miltiorrhiza were dried and crushed, and 0.1 g of the sample powder was mixed with 5 mL of distilled water and ground in a motor. Then, the ground samples were centrifuged (3,000 rpm for 1.5 min) and the supernatant was collected and subjected to anthrone colorimetry to ascertain the contents of glucose and fructose (Ju et al., 2024).

Determination of active ingredients content

The root tissue of S. miltiorrhiza from each group was crushed and passed through a 50-mesh sieve (with a pore diameter of 0.300 mm). Then, 0.5 g of the crushed sample was mixed with an equal amount of methanol (70%) and stirred ultrasonically for 30 min. The filtrate was passed through a 0.45-μm microporous membrane (Biosharp, Beijing, China) for subsequent high performance liquid chromatography (HPLC) analysis (Ju et al., 2024). The ultrasonic cleaner was provided by Ningbo Scientz Biotechnology Co., Ltd., with the model of SB-5200DT. The analysis conditions of Agilent 1120 high performance liquid chromatograph were as follows: Compass C18 chromatographic column (4.6 mm × 250 mm, 5 μm); mobile phase: ultrapure water (containing 0.2% acetic acid) as water phase, acetonitrile as organic phase; gradient elution procedure: 0–25 min, 5–35% B; 25–30 min, 35–55% B; 30–40 min, 55–75% B; 40–50 min, 75–95% B; flow rat: 1.0 ml/min; column temperature: 25 °C; injection volume: 10 μL (Ju et al., 2024).

Rhizosphere soil microbial amplicon sequencing

DNA extraction and PCR amplification

Total genome DNA from samples was extracted using CTAB/SDS method. DNA concentration and purity was monitored on 1% agarose gels (Barbier et al., 2019). DNA was diluted to 1 μg/μL using sterile water. The primer sequences for PCR are shown in Table 2. All PCR reactions were carried out with 15 μL of Phusion® High-Fidelity PCR Master Mix (New England Biolabs, Ipswich, MA, USA); 0.2 μM of forward and reverse primers, and about 10 ng template DNA (Ju et al., 2024). Thermal cycling is initial denaturation at 98 °C for 1 min, 30 cycles (98 °C for 10 s, 50 °C for 30 s, 72 °C for 30 s), and finally extension at 72 °C for 5 min (Ju et al., 2024).

Table 2 PCR amplification region and primer sequence.

Group	Amplification region	Primer sequence	
Bacteria	16Sv4	F:GTGCCAGCMGCCGCGGTAA
R:GGACTACHVGGGTWTCTAAT	
Fungi	ITS1-5F	F:GGAAGTAAAAGTCGTAACAAGG
R:GCTGCGTTCTTCATCGATGC	

Library construction and sequencing

Sequencing libraries were generated using TruSeq® DNA PCR-Free Sample Preparation Kit (Illumina, San Diego, CA, USA) following manufacturer’s recommendations and index codes were added (Monsigny, Petit & Roche, 1988). The library quality was assessed on the Qubit@2.0 Fluorometer (Thermo Fisher Scientific, Waltham, MA, USA) and Agilent Bioanalyzer 2100 system (Agilent, Santa Clara, CA, USA) (Ju et al., 2024). At last, the library was sequenced on an Illumina NovaSeq platform and 250 bp paired-end reads were generated.

Sequencing data processing

Paired-end reads were merged using FLASH (VI.2.7, http://ccb.jhu.edu/software/FLASH/); the splicing sequences were called raw tags. Quality filtering on the raw tags were performed under specific filtering conditions to obtain the high-quality clean tag according to the QIIME (V1.9.1, https://qiime.org/) quality controlled process. The tags were compared with the reference database (Silva database, https://www.arb-silva.de/) using UCHIME algorithm (UCHIME Algorithm, https://www.drive5.com/usearch/manual/uchime_algo.html) to detect chimera sequences, and then the chimera sequences were removed. Then the Effective Tags were finally obtained (Ju et al., 2024).

OTU cluster and species annotation

Sequences analysis were performed by Uparse software (Uparse v7.0.1001, http://drive5.com/uparse/). Sequences with ≥97% similarity were assigned to the same OTUs. 16S: The Silva Database (http://www.arb-silva.de/) was used based on Mothur algorithm to annotate taxonomic information. ITS: The Unite Database (https://unite.ut.ee/) was used based on blast algorithm to annotate taxonomic information (Ju et al., 2024). Finally, the data of each sample is homogenized. The sample with the smallest data volume is selected as the reference standard for homogenization processing. Subsequently, a Venn diagram is created (Ju et al., 2024).

Alpha diversity

Alpha diversity is applied in analyzing complexity of species diversity for a sample through six indices, including Observed-species, Chao1, Shannon, Simpson, ACE, Good-coverage. All this indices in our samples were calculated with QIIME (Version 1.7.0) and displayed with R software (Version 2.15.3; R Core Team, 2013) (Ju et al., 2024).

Beta diversity

Use Qiime software (Version 1.9.1) to calculate Unifrac distance and build UPGMA sample cluster tree. Principal coordinates analysis (PCoA) analysis was displayed by ade4 package and ggplot2 package in R software (R Core Team, 2013) (Monsigny, Petit & Roche, 1988). LEfSe (LDA Effect Size) analysis was performed using the LEfSe software, and the default setting of the LDA Score screening value was 4. Simper analysis using R software vegan package simper function. Based on FunGuild tool, the corresponding microbial ecological function classification was labeled (Monsigny, Petit & Roche, 1988).

Results

Effect of monocropping soil on the growth of S. miltiorrhiza

As shown in Table 3, the fresh and dry weights of the aboveground and underground plant parts of the monocropping soil group were reduced, when compared with those of the non-monocropping soil group. The biomass of the aboveground parts decreased by 27.1% and 30.2% respectively, while that of the underground parts decreased by 25.3% and 25.2%. Furthermore, compared with the monocropping soil group, the non-monocropping soil group had a greater number of normal leaves and a larger leaf area, a lower rate of withered leaves, and a similar leaf length-width ratio. Moreover, the underground part of the S. miltiorrhiza stubble was more developed, with a significantly increased length of the longest root, and slightly but not significantly increased taproot diameter (p = 0.19) and number of lateral roots (p = 0.07). These results indicated that monocropping soil conditions were not conducive to the morphological growth and biomass accumulation of both the aboveground and underground parts of S. miltiorrhiza, and exerted a significant impact on the overall yield of this plant.

Table 3 Effect of monocropping on growth index of S. miltiorrhiza.

Morphological index	NS-group	MS-group	
Aboveground fresh weight (g/plant)	4.76 ± 1.17	3.47 ± 0.90	
Aboveground dry weight (g/plant)	0.86 ± 0.16	0.60 ± 0.07*	
Fresh weight of underground part (g/plant)	4.78 ± 1.79	3.57 ± 1.25	
Dry weight of underground part (g/plant)	1.27 ± 0.48	0.95 ± 0.50	
Drying rate %	26.75 ± 4.32	24.75 ± 5.85	
Number of normal leaves (pieces)	31.0 ± 3.67	26.0 ± 7.87	
Number of dead leaves (pieces)	7.5 ± 3.20	12.5 ± 4.15	
Total number of leaves (pieces)	38.5 ± 1.66	38.5 ± 4.50	
Leaf area (cm2)	217.60 ± 37.94	183.76 ± 47.86	
Leaf length/leaf width	1.33~1.93	1.22~1.77	
The longest root length (cm)	16.00 ± 2.45	14.75 ± 2.59	
Main root diameter (cm)	0.63 ± 0.13	0.50 ± 0.07	
Number of root (article)	5.75 ± 0.43	4.25 ± 1.09	
Note:

* p < 0.05.

Differences in the physiological indices

Differences in the chlorophyll, soluble sugar, and sucrose contents

The contents of chlorophyll a, chlorophyll b, total chlorophyll, and carotenoids in the MS group exhibited a highly significant decrease (p < 0.001), with reductions of 42.4%, 41.5%, 42.1%, and 37.0%, respectively, when compared with those in the NS group (Fig. 1).

Figure 1 Difference of chlorophyll content between monocropping and non-monocropping S. miltiorrhiza.

The pigment content is expressed as mg/g (milligrams per gram of fresh tissue); ***p < 0.001.

As shown in Fig. 2, monocropping soil significantly affected the sugar contents in leaves and roots of S. miltiorrhiza compared with non-monocropping soil. In leaves, compared with the NS group, the contents of soluble sugar (ss) and sucrose (suc) in the MS group were significantly decreased (by 65.4% and 45.0%, respectively, p < 0.05). Although the contents of glucose and fructose showed a decreasing trend, the differences were not statistically significant (p > 0.05). In roots, the soluble sugar content in the MS group was also significantly reduced (with a decrease of 59.9%, p < 0.05). However, there were no significant differences in the contents of sucrose and fructose between the two groups (p > 0.05).

Figure 2 Difference of sugar content in roots and leaves of salvia miltiorrhiza in continuous and non-monocropping.

R stands for root, L stands for leaf, ss stands for soluble sugar, suc stands for sucrose, glu stands for glucose, fru stands for fructose. Note: *p < 0.05.

Difference in the active compounds content

The pharmacodynamic material basis of S. miltiorrhiza mainly includes water-soluble phenolic acid components (such as salvianolic acid B, rosmarinic acid) and fat-soluble tanshinone components (such as tanshinone IIA, cryptotanshinone, dihydrotanshinone I). These two types of components originate from different secondary metabolic pathways, and their contents are the core indicators for evaluating the quality of S. miltiorrhiza medicinal materials specified in the Pharmacopoeia of the People’s Republic of China. This study selected these key active components for analysis, aiming to comprehensively evaluate the potential impact of continuous cropping soil on the medicinal quality of S. miltiorrhiza from different metabolic pathways. It can be seen from Table 4 that continuous cropping soil has a specific impact on the content of active components in S. miltiorrhiza. Compared with NS group, MS group increased the content of salvianolic acid B by 28.8%, while the content of rosmarinic acid decreased by 51.4%. Among the tanshinone components, the contents of tanshinone I, tanshinone IIA and dihydrotanshinone I decreased by 55.4%, 4.4% and 100% respectively (the latter was difficult to detect in the roots under continuous cropping), while the content of cryptotanshinone increased by 50.0%.

Table 4 The content of effective components in S. miltiorrhiza (mg/g).

Active principle	NS	MS	
Tanshinone I	0.148 ± 0.002	0.066 ± 0.0001***	
TanshinoneIIA	1.137 ± 0.005	1.087 ± 0.005**	
Cryptotanshinone	0.008 ± 0.0004	0.012 ± 0.0002**	
Dihydrotanshinone I	0.075 ± 0.002	0***	
Salvianolic acid B	12.5 ± 0.082	16.1 ± 0.170***	
Rosmarinic acid	5.383 ± 0.045	2.617 ± 0.012***	
Note:

The content of dihydrotanshinone in group MS was extremely low and did not reach the lowest value of detection.

** Significant at 0.01 level

*** Significant at 0.001 level

Analysis of amplicon sequencing results

Sequencing data statistics and OTU analysis

By analyzing the sequencing results of bacteria and fungi in the rhizosphere soil of S. miltiorrhiza, 83,256, 82,089, 75,428 and 74,409 original tags, 77,993, 76,756, 74,600 and 73,511 active fragments were obtained for group NS and group MS. The effective rates were 92.52%, 92.32%, 89.00% and 86.85%, respectively, and the number of OUT was 3,942, 4,078, 980 and 1,017 species, respectively (Table S1). Subsequently, the representative OTU sequences obtained from groups NS and MS were subjected to cluster analysis, and the sequences were annotated at six different taxonomic levels: phyla, class, order, family, genus and species (Table S2).

α-diversity analysis

We conducted statistical analysis of α-diversity indices on the richness and evenness of the rhizosphere soil microbial community. For the bacterial community, no significant differences were observed between the continuous cropping soil (group MS) and non-continuous cropping soil (group NS) in all indices (including Shannon, Simpson, Chao1, Goods-coverage, and PD-whole-tree) (all p > 0.05). This indicates that under the conditions of this experiment, the continuous cropping history of the soil did not significantly change the overall richness and diversity of the bacterial community.

In contrast, the fungal community underwent significant changes. The Shannon index in continuous cropping soil (5.56 ± 0.21) was significantly higher than that in non-continuous cropping soil (5.12 ± 0.15) (p < 0.01). Similarly, the Simpson index in the continuous cropping group (0.93 ± 0.02) also significantly increased compared with the non-continuous cropping group (0.90 ± 0.02) (p < 0.05). However, the Chao1 index, which reflects species richness, and the PD-whole-tree index, which reflects phylogenetic diversity, did not show significant differences (p > 0.05). These results indicate that continuous cropping soil mainly improved the evenness and overall diversity of the fungal community, but did not significantly affect the total number of fungal species (richness) (Table 5).

Table 5 Alpha diversity.

Microbial type	Bacteria	Fungi	
Treatment	NS group	MS group	NS group	MS group	
Shannon	9.94 ± 0.48	9.85 ± 0.62	5.12 ± 0.15	5.56 ± 0.21**	
Simpson	1.00 ± 0.00	1.00 ± 0.00	0.90 ± 0.02	0.93 ± 0.02*	
Chao1	4,298.82 ± 353.38	4,376.41 ± 410.23	1,237.13 ± 282.73	1,254.23 ± 261.95	
Goods-coverage	0.99 ± 0.00	0.99 ± 0.00	0.99 ± 0.00	0.99 ± 0.00	
PD-whole-tree	241.02 ± 15.94	255.22 ± 21.91	507.46 ± 130.50	485.54 ± 110.98	
Note:

Independent samples t-test: * p < 0.05, ** p < 0.01. Shannon: The total number of categories in the sample and their proportions. The higher the community diversity, the more uniform the species distribution, and the larger the Shannon index. Chao1: Estimation of the total number of species in community samples. Coverage of goods: Coverage. The higher the coverage of the sort, the larger the index. PD whole tree: the genetic relationship of species in the community. Simpson: Diversity and uniformity of species distribution in a community. The analysis used the Simpson diversity index (1-D). The higher the species evenness, the greater the Simpson index.

β-diversity analysis

The bacterial composition notably varied between continuous and non-monocropping soils. Principal coordinate analysis showed the differences in the soil bacterial communities between groups NS and MS (Fig. 3A). The first and second principal component axes (contribution: 22.67% and 12.71%, respectively) distinguished the soil bacterial communities between monocropping and non-monocropping groups. In addition, the first and second principal component axes (contribution: 44.01% and 28.54%, respectively) also exhibited differences in the soil fungal community composition between groups NS and MS (Fig. 3B).

Figure 3 Distance PCoA analysis based on weighted unifrac: (A) bacteria, (B) fungus.

Analysis of the rhizosphere soil bacterial community structure

To evaluate the structural changes in the bacterial community, we analyzed the relative abundance of taxa at different taxonomic levels. Although the overall community composition of monocropping and non-monocropping soils appeared similar at higher taxonomic ranks (such as phylum, class, and order) (Figs. 4A–4C), significant changes were observed at more refined taxonomic levels. At the family level (Fig. 5A), the relative abundance of Lachnospiracea and Nitrosopharacea, both of which exhibit soil nitrogen cycle function, decreased by 38.32% and 22.75%, respectively, in the rhizosphere soil of the monocropping group. The relative abundances of Erysipelotrichaceae and Ruminococcaceae decreased by 42.49% and 40.82%, respectively, in the rhizosphere soil of the monocropping group. Concurrently, Staphylococcaceae, Peptostreptococcaceae, Lactobacillaceae, and Acidithiobacillaceae also exhibited varying degrees of decrease. At the genus level (Fig. 5B), the dominant genera in the rhizosphere soil bacterial community of groups NS and MS displayed alterations following monocropping. Specifically, the relative abundances of Staphylococcus (0.3% and 2.2%, respectively), Romboutsia (0.6% and 2.1%, respectively), Acidithiobacillus (0.006% and 1.1%), Leptospirillum (0.0006% and 1.0%, respectively), Allobaculum (0.9% and 0.5%), Blautia (1.3% and 0.8%), Lactobacillus (0.6% and 0.8%), Pseudomonas (1.4% and 1.0%), Ruminococcus (0.9% and 0.4%), and Bacillus (0.8% and 1.6%) notably differed between groups NS and MS.

Figure 4 Relative abundance map at different classification levels.

(A) bacterial phylum level, (B) bacterial class level, (C) bacterial order level, (D) fungal phylum level, (E) fungal class level, (F) fungal order level.

Figure 5 Contribution of bacterial community differences based on Simper analysis.

(A) Family level, (B) genus level note: The vertical axis represents the species, the horizontal axis is the sample, the bubble size represents the relative abundance of the species, and ‘Contribution’ is the contribution of the species to the difference between the two groups.

Analysis of the rhizosphere soil fungal community structure

Venn diagrams showing the overlap of OTUs between groups are provided in Figs. S1 and S2. Furthermore, marked differences in the soil fungal community at the phylum level were observed between the groups NS and MS (Fig. 4D), and the relative abundances of the top 10 dominant phyla, including Mortiellomycota, Chytridillomycota, Zoopagomycota, Aphelidiomycota, Mucoromycota, Glomeromycota, and Blastocladiomycota, increased by 127.5%, 325.0%, 32.4%, 253.1%, 265.7%, 14.0%, and 1230.8%, respectively, while those of Ascomycota, Basidiomycota, and Rozellomycota decreased by 19.42%, 90.6%, and 30.9%, after monocropping.

As shown in Fig. 4E, the dominant fungal classes in groups NS and MS were Sordariomycetes (37.2% and 38.7%), Mortierellomycetes (9.1% and 20.7%), Agaricomycetes (5.8% and 4.4%), Orbiliomycetes (10.4% and 6.1%), Dothideomycetes (10.2% and 4.0%), Eurotiomycetes (7.4% and 4.9%), Leotiomycetes (2.3% and 1.2%), and Tremellomycetes (1.7% and 2.6%). Furthermore, the dominant fungal orders in groups NS and MS were Pyrrophyta (23.7% and 16.6%), Mortierela (9.1% and 20.7%), Thesephorales (4.6% and 0.08%,), Orbiliales (10.4% and 6.1%), Microscales (3.0% and 9.2%), Capnodiales (8.0% and 1.1%), Eurotiales (8.0% and 1.1%), Pleospores (7.0% and 4.6%), and Sordariales (2.0% and 2.7%) (Fig. 4F). The cluster heatmap of species abundance (Fig. 6) revealed significant differences in species richness and composition at the family level between groups NS and MS. The proportion of Mortieaceae and Cysaceae in the rhizosphere fungal community of monocropping S. miltiorrhiza increased, while the proportion of Coniochaetacaeae decreased.

Figure 6 Clustering heat map of relative abundance of soil fungi in continuous and non-continuous cultivation of Salvia miltiorrhiza at family classification level.

Certain common soil-borne pathogenic fungi, including Leptosphaeria turcica, Cladosporium species, Alternaria, Fusarium solani, and other Fusarium subspecies, became more abundant (Tables S3 and S4). Based on LEfSe (LDA Effect Size) analysis, the relationship between monocropping and non-monocropping soils was further explored. The results revealed that the abundance of the microbial species in the S. miltiorrhiza soil was significantly different before and after monocropping. A total of 30 biomarkers with LDA score > 4 were enriched in the two groups (Fig. 7A), with the abundance of Mortierella, Lophotrichus, Conocybe, Aspergillus, Arthrobotrys, and Cladosporium being statistically different in the two groups (Fig. 7B). Mortierella polycephala, Lophotrichus sp., Mortierella stylospora, and Conocybe pubescens were the biomarkers in the monocropping soil, while Mortierella alpina, Arthrobotrys amerospora, and Cladosporium sp. were the biomarkers in the non-monocropping soil.

Figure 7 LEfSe analysis of rhizosphere soil microorganisms in monocropping and non-monocropping S. miltiorrhiza.

(A) LDA value distribution histogram, (B) evolutionary branch diagram.

Function prediction analysis

As shown in Figs. 8A and 8B, the pathotrophic functions of plants and animals were relatively dominant in the soil bacterial community. The abundance of saprophytic bacteria was relatively low, showing only a slight difference between groups NS and MS. These results showed that the functions of bacteria in the rhizosphere soil of S. miltiorrhiza in group NS and group MS were similar. The abundance of internal parasitic nutritional fungi in the soil fungal community decreased, while that of saprophytic nutritional fungi significantly increased in group NS, which may mean that the symbiotic relationship between soil fungi and plants was destroyed and the growth of S. miltiorrhiza was adversely affected. Subsequently, principal component analysis was performed on the abundance statistics results based on the database functional annotations (Fig. 8C). The soil fungal functional abundance showed good separation. The first and second principal component axes (contribution: 24.3% and 18.76%, respectively) distinguished the soil fungal composition between groups NS and MS. In contrast, the soil bacterial functional abundance between the two groups did not exhibit good separation (Fig. 8D). Therefore, it can be speculated that the functions of bacteria and fungi in the rhizosphere soil of S. miltiorrhiza treated with continuous cropping soil promoted the enrichment of pathogenic fungi to a certain extent compared with non-continuous cropping soil.

Figure 8 NS group and MS group function annotation relative abundance display.

(A) Bacterial FunGuild function annotation relative abundance histogram, (B) fungal FunGuild function annotation relative abundance histogram, (C) fungal FunGuild function annotation principal component analysis (PCA) result display, (D) bacterial FunGuild function annotation PCA result display.

Correlation analysis between soil microbial species abundance and S. miltiorrhiza growth indexes

The S. miltiorrhiza root morphology and root biomass accumulation were significantly positively correlated with the relative abundances of Lecythophora, Aspergillus, Alternaria, Eurotium, and other fungi, and the bacterium Pseudomonas, and were negatively correlated with Rhizophlycti and Papulaspora. The contents of chlorophyll a, chlorophyll b, and carotenoids in S. miltiorrhiza were notably positively correlated with the relative abundances of Eurotium, Macroventuria, Gibberella, Geomyces, Aspergillus, Cladosporium, Arthrobotrys, Solirubrobacter, Clostridium innocuum, Lysobacter, and Gaiella, but markedly negatively correlated with the relative abundances of fungi such as Papulaspora, Fusicolla, Chaetomium, Solicoccozyma, and Mortierella, and some bacteria such as Ohtaekwangia, Streptomyces, Bacillus, and Leptospirillum (Figs. 9A and 9B).

Figure 9 Spearman analysis of bacterial and fungal species abundance and physiological indexes of S. miltiorrhiza.

FWA: Aboveground fresh weight; FWR: Root fresh weight; LRL: longest root length; TD: taproot diameter; CA: Chlorophyll A; CB: Chlorophyll B; CAN: Carotenoids. *is significant at 0.05 level.

Discussion

The experimental design of this study involved utilizing soils collected from fields with a history of continuous cropping and those without for cultivating S. miltiorrhiza seedlings in pots. The observed variations in the growth, physiology, quality, and rhizosphere microorganisms of S. miltiorrhiza primarily reflect inherent disparities between the two soil types established prior to planting, influenced by historical cropping practices, and their effects on the growth of S. miltiorrhiza during the current planting cycle. While this design effectively isolates soil factors and controls environmental variables, the absence of baseline data on the physical and chemical properties and microbial communities of the soil before planting hinders precise quantification of the contributions of “inherent differences” due to continuous cropping history vs. plant-mediated effects on the observed phenotypic disparities in S. miltiorrhiza seedlings and changes in the rhizosphere microbial community. It does not replicate the dynamic cumulative impacts of plant-soil-microorganism interactions in the actual continuous cropping process in the field, presenting a limitation of this study. Nonetheless, “continuous cropping soil” itself embodies the material substrate and ultimate reflection of continuous cropping challenges. Elucidating plant physiological responses in such challenging soils, alterations in quality metabolic pathways, and the distinct structure of the rhizosphere microbial community holds direct implications for understanding the mechanisms of continuous cropping challenges, identifying early-diagnosis biomarkers, and devising soil enhancement strategies.

Monocropping soil profoundly impacts the morphological and physiological characteristics of S. miltiorrhiza, reflecting plant growth status through indicators such as root and leaf development, carbohydrate and active ingredient levels. Leaf number and area influence plant photosynthesis and crop yield. Root morphology and vitality directly impact nutrient absorption and photosynthetic product transport efficiency, thereby affecting S. miltiorrhiza growth (Liu et al., 2020; Li & Zhang, 2015). In this study, a pot experiment was conducted with S. miltiorrhiza, revealing decreased biomass, dried aboveground leaves, reduced effective leaf area, and shorter, narrower underground main roots in continuously cropped soil. Monocropping significantly affects S. miltiorrhiza’s physiological indices, leading to decreased total chlorophyll content, reduced soluble sugar accumulation (sucrose, glucose, fructose), increased salvianolic acid B content, and decreased rosmarinic acid content in water-soluble components. Additionally, tanshinone I, tanshinone IIA, and dihydrotanshinone I contents decrease post-monocropping, except for cryptotanshinone.

The diversity of soil microorganisms serves as a comprehensive indicator of soil microflora changes and soil health (Qin et al., 2017). For example, Wu et al. (2017) observed an imbalance and structural disorder in the rhizosphere microflora of Pseudostellaria heterophylla under continuous monoculture, potentially due to phenolic acids. Similarly (Wu et al., 2017), Yu, Gao & Sun (2022) noted an increase in harmful microorganisms in the rhizosphere soil of Asarum after monocropping, disrupting the balance between beneficial and harmful bacteria. In a pot experiment, our study revealed a decrease in bacterial community diversity but an increase in fungal community diversity in the rhizosphere soil of S. miltiorrhiza. Furthermore, compared to non-continuous cropping soil, the proportion of bacteria and fungi in the rhizosphere of continuously cropped S. miltiorrhiza was lower, with an enrichment of pathogenic fungi like Alternaria subsp. and Fusarium solani. Previous research has highlighted the significant impact of cropping systems on the soil bacteria-to-fungi ratio (Lauber et al., 2008; Pereira e Silva et al., 2012). The rhizosphere micro-ecosystem, as an organic integration of plants, soil, and microorganisms, plays a crucial role in plant-environment interactions and functions (Kuypers, Marchant & Kartal, 2018), closely linked to the challenges of monocropping. The relationship between microbial community biodiversity and community stability is generally positive yet intricate and context-dependent. The observed reduction in rhizosphere bacterial richness and diversity, along with the increase in harmful fungi abundance, may contribute to the suboptimal growth of S. miltiorrhiza. Our findings support this hypothesis. Spearman analysis revealed a significant correlation between rhizosphere microbial population structure and the root development and photosynthetic intensity of S. miltiorrhiza under monocropping soil, suggesting a potential link between microbial population changes and nutrient transport and plant growth. Further research is warranted to elucidate the underlying mechanisms.

In continuous cropping soil, Actinobacteria and Bacteroidetes decreased, while Firmicutes and other oligotrophic bacteria increased, indicating alterations in the rhizosphere soil environment prompting certain soil microorganisms to adjust their metabolic pathways to new conditions. Previous studies have shown Mortierella’s ability to infect plants, causing widespread crop infections. Our research observed a decrease in Nitrosophaeraceae abundance at the family level, key players in the soil nitrogen cycle, alongside an increase in pathogenic Mortierellaceae and Cystaceae. These findings underscore the impact of monocropping on soil bacterial community composition. The shift in soil bacterial community diversity post-monocropping, characterized by decreased beneficial bacteria and increased pathogenic bacteria, may disrupt rhizosphere soil microflora function, exacerbating monocropping issues. Analysis of diversity indices revealed marked changes in soil fungal community structure, with increased richness and diversity post-monocropping. Genus-level cluster analysis indicated notable increases in Alternaria and Fusarium, important plant pathogenic fungi, with pathogenic fungi like F. solani enriched in S. miltiorrhiza rhizosphere soil under monocropping, potentially contributing to worsened disease incidence. The rise of pathogenic fungi as the dominant group can upset rhizosphere soil microecology balance, compromising growth conditions.

Current research on the challenges of monocropping medicinal plants primarily examines soil degradation, allelopathic autotoxicity, imbalances in rhizosphere microecology, and increased soil-borne diseases. Understanding monocropping obstacles necessitates a focus on the dynamic changes and functional analysis of rhizosphere microbial communities associated with medicinal plants. Competition for rhizosphere ecological niches between beneficial microbial groups and pathogenic fungi, influenced by root exudates, significantly impacts the growth and metabolism of medicinal plants (Dong et al., 2017). Prolonged monoculture results in the accumulation of acidic root exudates, such as phenolic acids and coumarins, attracting pathogenic fungi to the rhizosphere. Root-secreted components like coumaric acid, benzoic acid, and saponins from Panax notoginseng and Panax ginseng roots exhibit strong chemotaxis towards pathogens, leading to increased pathogen abundance in the rhizosphere over time (Bao et al., 2022; Zhang & Lin, 2009). The acidified soil environment favors the colonization and growth of soil fungal communities, shifting the monocropped soil from a bacterial to a fungal-dominated type, negatively impacting the host plant’s growth and development (Wang et al., 2021; Dong et al., 2018; Jin, Wu & Zhou, 2020; Hontoria et al., 2019). However, existing research, including the present study, predominantly focuses on unilateral changes in plants and the rhizosphere microenvironment post-monocropping, overlooking the intricate interactions among plants, root exudates, and rhizosphere microorganisms. Future studies should investigate the reciprocal exchanges between plant roots and soil rhizosphere microorganisms, elucidating the factors influencing rhizosphere soil microbial communities.

The findings of this study vividly illustrate the stress-induced and adaptive changes in S. miltiorrhiza seedlings from physiology to rhizosphere microecology in a soil environment characterized by established continuous cropping challenges. Subsequent research should integrate long-term field trials or continuous multi-season pot simulations to dynamically monitor the progressive effects of the continuous cropping process on soil properties and plant vitality, aiming to comprehensively unveil the formation mechanism of continuous cropping challenges in S. miltiorrhiza.

Conclusion

In summary, continuous cropping soil significantly affects the morphology, physiological activity, and biomass accumulation of S. miltiorrhiza, leading to changes in the community structure and metabolic functions of bacteria and fungi in the rhizosphere soil. Compared with non-continuous cropping soil, the community composition of bacteria and fungi in continuous cropping soil has also changed, and pathogenic fungi such as Fusarium solani have accumulated. These changes may further lead to a decline in the quality of S. miltiorrhiza. In summary, the present study systematically elucidated the effects of monocropping soil on the growth and development of S. miltiorrhiza and rhizosphere microflora, and discussed the mechanisms of monocropping obstacles and quality decline of a Chinese medicinal plant based on rhizosphere microorganisms. The results obtained are crucial for regulating rhizosphere soil microecology and improving the cultivation of Chinese medicinal plants. Future research should dynamically track the gradual effect of continuous cropping process itself on rhizosphere soil microorganisms, and provide a theoretical basis for formulating strategies to alleviate continuous cropping obstacles.

Supplemental Information

Supplemental Information 1 OTU Wayne diagram of bacteria.

Supplemental Information 2 OTU Wayne diagram of fungi.

Supplemental Information 3 Sequencing data statistics of bacteria and fungi in rhizosphere soil of continuous and non-continuous cropping of Salvia miltiorrhiza.

Note: Original tags = tag sequences obtained by splicing; effective label = the label sequence finally used for subsequent analysis after filtering the chimera; base = the base of the final valid data; effectiveness (%) = the percentage of the number of valid labels to the number of original PEs; oTUs = the number of operational taxonomic units.

Supplemental Information 4 OTU statistics of bacterial and fungal communities in rhizosphere soil of continuous and non-monocropping of S. miltiorrhiza.

Supplemental Information 5 Relative abundance of fungi (genus level).

Supplemental Information 6 Relative abundance of fungi (species level).

Additional Information and Declarations

Competing Interests

The authors declare that they have no competing interests.

Author Contributions

Ji Dong Ju conceived and designed the experiments, performed the experiments, prepared figures and/or tables, authored or reviewed drafts of the article, and approved the final draft.

Yuan Meng conceived and designed the experiments, performed the experiments, analyzed the data, prepared figures and/or tables, authored or reviewed drafts of the article, and approved the final draft.

Bingqian Zhou conceived and designed the experiments, analyzed the data, prepared figures and/or tables, authored or reviewed drafts of the article, and approved the final draft.

Guohong Yang performed the experiments, authored or reviewed drafts of the article, and approved the final draft.

Xinyu Fu performed the experiments, prepared figures and/or tables, and approved the final draft.

Wei Liu conceived and designed the experiments, performed the experiments, analyzed the data, prepared figures and/or tables, authored or reviewed drafts of the article, and approved the final draft.

Data Availability

The following information was supplied regarding data availability:

The sequence data are available at the NCBl ShortRead Archive: PRJNA1129673, PRJNA1129675.

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
