# Peer review of "Effects of monocropping soil on plant growth and rhizosphere microbial community structure of Salvia miltiorrhiza Bge"

_PeerJ, doi:10.7717/peerj.20379_

## Round 0.1 · original submission · Major Revisions

· Academic Editor

Major Revisions

Please address all the concerns raised by the reviewers.

Reviewer 1 ·

Basic reporting

English need to be improved from native speaker.

Experimental design

no

Validity of the findings

no

Additional comments

Thank you for submitting the manuscript titled “Effects of monocropping soil on plant growth and rhizosphere microbial community structure of Salvia miltiorrhiza Bge”. The study provides valuable insights into monocropping-induced changes in Salvia miltiorrhiza physiology and rhizosphere microbial communities. The experimental design is rigorous, and the integration of plant physiological data with microbiome analysis is commendable. However, several issues require clarification and improvement to enhance scientific readability.
Please also see pdf file for comments.
1. Figs. 1–3, Table 3. Clarify which statistical tests were used (ANOVA, t-test, etc.) and include p-values or significance annotations.
2. Lines 198–203. Clarify whether increases/decreases in metabolite levels are statistically significant.
3. Restrict "significant" to statistical results only; otherwise use "notable" or "marked."
4. Numerous grammar/spelling/formatting issues. Lines 77, 128, 131, 134, 138, 248, etc.
5. Lines 14–15, 319–321. Simplify complex sentences.
6. Lines 382–393. Add implications for sustainable cultivation and future research needs.
Lines 385–393 (Conclusion):
o The claim that fungal diversity "increased" (Shannon: 5.12 vs. 5.56) is statistically valid but ecologically minor. Change the language to reflect the modest change.
o The link between Fusarium solani enrichment and monocropping obstacles is correlational. Use causal language cautiously (e.g., "may aggravate" instead of "led to").
Lines 87–89:

The labels "Group f" and "Group l" are undefined. Specify what "f" and "l" represent (e.g., "f" = non-monocropping, "l" = monocropping).
Lines 175–184 (Results):

Table 3 reports significant differences in aboveground dry weight (p<0.05) but does not clarify statistical tests used (e.g., t-test, ANOVA). Specify the statistical method and adjust for multiple comparisons if applicable.

Line 182: The phrase "slightly but not significantly increased" (Line 182) is subjective. Replace with exact p-values or confidence intervals

Annotated reviews are not available for download in order to protect the identity of reviewers who chose to remain anonymous.

Reviewer 2 ·

Basic reporting

This study aims to investigate the impact of monocropping on Salvia miltiorrhiza (Danshen), focusing on the associated changes in rhizosphere microbial communities. Through pot experiments and high-throughput sequencing of bacterial and fungal markers, the authors show that plants grown in monocropping soil, present diiferent physiological and morphological indexes in S. miltiorrhiza, alongside potential shifts in rhizosphere microbial community structure, and an increase in potentially pathogenic fungi such as Fusarium solani, compared to plants grown in non-monocropping soil.

Despite the effort put into the experiments and the results obtained, I believe the manuscript would benefit from substantial revision for clarity and language. The current writing quality, particularly the English expression, often makes the text difficult to follow and may obscure the intended meaning in several sections. A thorough language editing would greatly enhance readability.

In addition, the framing of the study could be improved. At times, the manuscript appears to interpret results beyond the scope of what was directly tested. A clearer articulation of the research questions and hypotheses, along with more precise alignment between those questions and the experimental design, would strengthen the manuscript’s focus. I will provide more detailed suggestions below to support these revisions.

Experimental design

While the study aims to investigate the effects of monocropping on Salvia miltiorrhiza, the experimental setup uses soil sourced from monocropped and non-monocropped fields in pot experiments, rather than conducting actual in situ monocropping over time. Therefore, results primarily reflect pre-existing differences in soil properties rather than the direct effects of continuous monoculture per se. The authors should more clearly acknowledge this limitation and frame their interpretations accordingly, avoiding overgeneralization of the results as being solely attributable to monocropping effects.

Additionally, the authors do not provide sufficient detail regarding the management history of the soils classified as monocropped and non-monocropped. It would be important to clarify how these soils were managed prior to the experiment, including the duration of monocropping, whether the fields were adjacent or from distinct locations, and any other relevant agronomic practices. This contextual information is essential for properly interpreting the results and assessing the comparability of the soil treatments.

Because of the nature of the experiments, a thorough baseline characterization of both soil types is essential since is the primary variable distinguishing the treatments. The study should include physicochemical parameters and microbial community profiles prior to planting. Such comparative data would help disentangle the influence of initial soil conditions from the plant-mediated effects observed during the pot experiment. Without this information, it is difficult to assess the extent to which the reported plant and microbial outcomes result from monocropping history versus inherent soil differences.

Validity of the findings

While the study addresses a relevant question regarding the impact of monocropping on Salvia miltiorrhiza and its associated microbial communities, there are concerns about the experimental framing that limit the strength of the conclusions. In particular, the distinction between monocropping and non-monocropping conditions is not clearly established, as the experiment appears to rely on soil from different sources rather than a long-term field-based monocropping system. Without detailed information on the management history and baseline properties of the soils used, it is difficult to attribute observed effects solely to monocropping.

Furthermore, some of the key findings, such as differences in alpha diversity and microbial community composition, are not supported by appropriate statistical comparisons or sufficiently detailed analysis. Overall, while the data collected are potentially valuable, the current presentation and methodological gaps weaken the validity of the conclusions. Strengthening the experimental design rationale, clarifying treatment definitions, and incorporating proper statistical analyses would significantly improve the reliability of the findings.

Additional comments

The naming of experimental groups is somewhat unclear and should be reconsidered. The authors refer to plants grown in non-monocropping soil as "f" and those grown in monocropping soil as "l" This labeling is not intuitive and makes it difficult to follow the results and interpret the figures accurately. I strongly recommend renaming the groups using more descriptive terms such as "monocropping soil" and "non-monocropping soil" to improve clarity throughout the manuscript and figures.

In Figure 1, the y-axis is labeled "chlorophyll content (mg/g)," but this is misleading, as the third group of bars represents carotenoids, which are not chlorophyll compounds. The authors should revise the y-axis title to reflect the inclusion of multiple pigment types, or separate them clearly. Additionally, the units used ("mg/g") should be clearly explained in the figure legend to ensure proper interpretation.

In Figure 2, which presents sugar content in roots and leaves, the labeling of sample groups appears to contain a typographical error. The authors refer to "L" as representing leaves and "F" as roots, which conflicts with the earlier labeling of "L" and "F" as treatment groups. Given the context, it seems likely that "R" was intended to represent roots rather than "F." This should be corrected for clarity and consistency. Additionally, the text describing the figure would benefit from greater specificity regarding statistical differences. Please clearly state which comparisons showed significant differences and which did not, and ensure that this is consistent with the figure’s annotations (e.g., asterisks or other indicators). This will help readers better understand the biological relevance of the results.

In the "Differences in the active compounds content" section, it would be helpful for the authors to briefly explain the rationale for selecting these specific active compounds for analysis. While their measurement is clearly relevant to the quality assessment of S. miltiorrhiza, providing context on their pharmacological importance or their role as key bioactive markers would help readers understand why these particular compounds were prioritized and how they relate to the overall objective of evaluating monocropping effects on plant quality.

In the "Sequencing data statistics and OTU analysis" section, the content and structure of Table 5 are unclear. Several of the numerical values described in the text do not appear to be included in the table, making it difficult to follow or verify the results. I recommend clarifying the table to ensure that all referenced data are clearly presented and correspond with the descriptions in the text. Additionally, given the detailed and technical nature of the OTU statistics, it may be more appropriate to move this information to the supplementary material.

In the Alpha Diversity Analysis section, I recommend including appropriate statistical comparisons between treatment groups to determine whether the observed differences in alpha diversity are significant. While some diversity values appear lower in certain cases, formal statistical testing across multiple diversity indices is essential to support any conclusions regarding treatment effects. Additionally, the relative abundance plots should be presented as main figures rather than supplemental material, as they provide important context for interpreting microbial community structure. Notably, the bacterial community compositions at different taxonomic levels appear nearly identical between treatments. Based on the current data, there does not seem to be a meaningful difference in alpha diversity between the communities. This should be more clearly addressed in the analysis and discussion.

The statement, "This implies that monocropping altered the soil bacterial community OTU composition to some extent, yet the overall structure of the dominant taxa was relatively stable," may be misleading. OTU composition is known to vary even between replicate samples from the same soil or field, so differences in OTUs alone are not necessarily indicative of meaningful shifts in community structure. What is more relevant for interpretation is whether specific taxonomic groups or species were uniquely present, enriched, or depleted in one treatment versus the other.

In Figure 4, while different OTUs are observed between groups, it is unclear whether these correspond to distinct taxonomic ranks or functionally relevant shifts. Did the authors identify any taxa that were uniquely present in one condition? Without such context, it may be premature to conclude that community composition was altered but structurally stable. I recommend revisiting this conclusion and aligning it more closely with taxonomic-level analysis rather than OTU-level variability, which can be inherently noisy.

In Figure 5, the taxonomic profiles between treatment groups appear highly consistent, with no clear visual indication of substantial differences in the composition or relative abundance of major taxa. While differences in OTU presence are noted, these may reflect natural variability rather than meaningful ecological shifts. As OTUs can vary significantly even between technical or biological replicates, conclusions based solely on OTU differences may not be robust.

---

## Round 0.2 · Minor Revisions

· Academic Editor

Minor Revisions

Please address the remaining minor comments.

Reviewer 1 ·

Basic reporting

I am not able to decide about English quality but it needs minor correction in some sections.

Experimental design

The submission should clearly define the research question, which must be relevant and meaningful. The knowledge gap being investigated should be identified, and statements should be made as to how the study contributes to filling that gap.

Validity of the findings

Table 5 need statistical analysis for validation of significance difference.

Additional comments

The manuscript has been much improved and is likely to acceptable after addressing one minor and important issue.
The authors must give appropriate statistical analysis for the alpha diversity data (Table 5). Calculate and report the mean ± standard deviation for each index (Shannon, Chao1, etc.) for both groups from their five biological replicates. Use appropriate statistical test (e.g., t-test) to determine significant differences. The results, including any significance difference, must be updated in the paper and table.

---

## Round 0.3 · Major Revisions

· Academic Editor

Major Revisions

Please address the remaining minor comments by reviewer 1, and these comments from me:

Experimental design
While the study aims to investigate the effects of monocropping on Salvia miltiorrhiza, the experimental setup uses soil sourced from monocropped and non-monocropped fields in pot experiments, rather than conducting actual in situ monocropping over time. Therefore, results primarily reflect pre-existing differences in soil properties rather than the direct effects of continuous monoculture per se. The authors should more clearly acknowledge this limitation and frame their interpretations accordingly, avoiding overgeneralization of the results as being solely attributable to monocropping effects.

Additionally, the authors do not provide sufficient detail regarding the management history of the soils classified as monocropped and non-monocropped. It would be important to clarify how these soils were managed prior to the experiment, including the duration of monocropping, whether the fields were adjacent or from distinct locations, and any other relevant agronomic practices. This contextual information is essential for properly interpreting the results and assessing the comparability of the soil treatments.

Because of the nature of the experiments, a thorough baseline characterization of both soil types is essential since is the primary variable distinguishing the treatments. The study should include physicochemical parameters and microbial community profiles prior to planting. Such comparative data would help disentangle the influence of initial soil conditions from the plant-mediated effects observed during the pot experiment. Without this information, it is difficult to assess the extent to which the reported plant and microbial outcomes result from monocropping history versus inherent soil differences.

Validity of the findings
While the study addresses a relevant question regarding the impact of monocropping on Salvia miltiorrhiza and its associated microbial communities, there are concerns about the experimental framing that limit the strength of the conclusions. In particular, the distinction between monocropping and non-monocropping conditions is not clearly established, as the experiment appears to rely on soil from different sources rather than a long-term field-based monocropping system. Without detailed information on the management history and baseline properties of the soils used, it is difficult to attribute observed effects solely to monocropping.

Furthermore, some of the key findings, such as differences in alpha diversity and microbial community composition, are not supported by appropriate statistical comparisons or sufficiently detailed analysis. Overall, while the data collected are potentially valuable, the current presentation and methodological gaps weaken the validity of the conclusions. Strengthening the experimental design rationale, clarifying treatment definitions, and incorporating proper statistical analyses would significantly improve the reliability of the findings.
Additional comments
The naming of experimental groups is somewhat unclear and should be reconsidered. The authors refer to plants grown in non-monocropping soil as "f" and those grown in monocropping soil as "l" This labeling is not intuitive and makes it difficult to follow the results and interpret the figures accurately. I strongly recommend renaming the groups using more descriptive terms such as "monocropping soil" and "non-monocropping soil" to improve clarity throughout the manuscript and figures.

In Figure 1, the y-axis is labeled "chlorophyll content (mg/g)," but this is misleading, as the third group of bars represents carotenoids, which are not chlorophyll compounds. The authors should revise the y-axis title to reflect the inclusion of multiple pigment types, or separate them clearly. Additionally, the units used ("mg/g") should be clearly explained in the figure legend to ensure proper interpretation.

In Figure 2, which presents sugar content in roots and leaves, the labeling of sample groups appears to contain a typographical error. The authors refer to "L" as representing leaves and "F" as roots, which conflicts with the earlier labeling of "L" and "F" as treatment groups. Given the context, it seems likely that "R" was intended to represent roots rather than "F." This should be corrected for clarity and consistency. Additionally, the text describing the figure would benefit from greater specificity regarding statistical differences. Please clearly state which comparisons showed significant differences and which did not, and ensure that this is consistent with the figure’s annotations (e.g., asterisks or other indicators). This will help readers better understand the biological relevance of the results.

In the "Differences in the active compounds content" section, it would be helpful for the authors to briefly explain the rationale for selecting these specific active compounds for analysis. While their measurement is clearly relevant to the quality assessment of S. miltiorrhiza, providing context on their pharmacological importance or their role as key bioactive markers would help readers understand why these particular compounds were prioritized and how they relate to the overall objective of evaluating monocropping effects on plant quality.

In the "Sequencing data statistics and OTU analysis" section, the content and structure of Table 5 are unclear. Several of the numerical values described in the text do not appear to be included in the table, making it difficult to follow or verify the results. I recommend clarifying the table to ensure that all referenced data are clearly presented and correspond with the descriptions in the text. Additionally, given the detailed and technical nature of the OTU statistics, it may be more appropriate to move this information to the supplementary material.

In the Alpha Diversity Analysis section, I recommend including appropriate statistical comparisons between treatment groups to determine whether the observed differences in alpha diversity are significant. While some diversity values appear lower in certain cases, formal statistical testing across multiple diversity indices is essential to support any conclusions regarding treatment effects. Additionally, the relative abundance plots should be presented as main figures rather than supplemental material, as they provide important context for interpreting microbial community structure. Notably, the bacterial community compositions at different taxonomic levels appear nearly identical between treatments. Based on the current data, there does not seem to be a meaningful difference in alpha diversity between the communities. This should be more clearly addressed in the analysis and discussion.

The statement, "This implies that monocropping altered the soil bacterial community OTU composition to some extent, yet the overall structure of the dominant taxa was relatively stable," may be misleading. OTU composition is known to vary even between replicate samples from the same soil or field, so differences in OTUs alone are not necessarily indicative of meaningful shifts in community structure. What is more relevant for interpretation is whether specific taxonomic groups or species were uniquely present, enriched, or depleted in one treatment versus the other.

In Figure 4, while different OTUs are observed between groups, it is unclear whether these correspond to distinct taxonomic ranks or functionally relevant shifts. Did the authors identify any taxa that were uniquely present in one condition? Without such context, it may be premature to conclude that community composition was altered but structurally stable. I recommend revisiting this conclusion and aligning it more closely with taxonomic-level analysis rather than OTU-level variability, which can be inherently noisy.

In Figure 5, the taxonomic profiles between treatment groups appear highly consistent, with no clear visual indication of substantial differences in the composition or relative abundance of major taxa. While differences in OTU presence are noted, these may reflect natural variability rather than meaningful ecological shifts. As OTUs can vary significantly even between technical or biological replicates, conclusions based solely on OTU differences may not be robust.

Reviewer 1 ·

Basic reporting

No Comment

Experimental design

no comment

Validity of the findings

no comment

Additional comments

1. The legend for Figure 2 states "R stands for root, L stands for leaf". However, in the figure itself, the labels appear to be "R" and "F". Please correct the legend to reflect the actual labels used in the figure (likely "F" for leaf and "R" for root) to avoid confusion.

2. The Data Availability statement lists two BioProject accession numbers (PRJNA1129673, PRJNA1129675). Please specify the accession associated with the 16S bacterial data and the one corresponding to the ITS fungal for maximum clarity and ease of access for readers.
3. Minor variations exist in the formatting of the binomial nomenclature. Ensure consistent use of italics for all genus and species names throughout the article, including figures and tables.

4. Please double-check all references for consistent journal abbreviation style and ensure all DOI links are functional (e.g., References 5, 6, 7, 14, 19, 23 appear to be missing the https://doi.org/ prefix and may not be hyperlinked correctly).

---

## Round 0.4 · accepted · Accept

· Academic Editor

Accept

Thanks for addressing all comments!